# Identification and correction of snow depth bias in ERA5 datasets over Central Europe using machine learning

Gabiel Stachura<sup>1,2,3</sup>, Zbigniew Ustrnul<sup>2,3</sup>,

<sup>1</sup>Doctoral School of Exact and Natural Sciences, Jagiellonian University, Cracow, 31-007, Poland <sup>2</sup>Department of Climatology, Institute of Geography and Spatial Management, Jagiellonian University, Cracow, 31-007 Poland <sup>3</sup>Institute of Meteorology and Water Management – National Research Institute, Cracow, 30-215, Poland *Correspondence to*: Gabriel Stachura (gabriel.stachura@imgw.pl)

Abstract. Accurate estimation of snow depth is a crucial problem from both meteorological and hydrological points of view.

Global and regional reanalyses still struggle to address it, mostly because the scale of snow spatial heterogeneity is widely beyond current resolutions of the databases. In the study, snow depth estimation from Copernicus reanalyses ERA5 and ERA5-Land are compared and evaluated against point measurements in Poland, Czech Republic and Slovakia in winter seasons 2001/2002 - 2020/2021. Additionally, a Random Forests (RF) model is developed to reduce identified errors based on various environmental variables and parameters derived from the reanalyses and a digital elevation model. As mountains are main snow water reservoirs for Central Europe, the model is then used to spatially downscale snow depth over a fine-scaled subdomain in mountainous terrain. For both reanalyses, the deviations are relatively small in flat or gently rolling terrain (below 500 m a.s.l.). ERA5 (0.25°) outperforms ERA5-Land (0.1°) due to the presence of data assimilation. Since only synop measurements are assimilated, errors are the lowest for these stations, however, lower-ranked stations are also affected. In more complex terrain, both reanalyses exhibit an underestimation of snow that increases with elevation. In this area, ERA5-Land is slightly less biased due to its higher resolution and the fact that observations from mountainous sites are often masked out from the data assimilation in ERA5. The proposed RF model improves accuracy of estimation by around 48% with respect to the best reanalysis. The results of spatial downscaling certainly provide added value to the problem of snow estimation in complex terrain. Although they cannot be considered entirely valid and reliable since not all factors determining spatial variability of snow at such resolution are taken into account, they might be useful for future studies concerning, e.g., climatological variability of snow with respect to altitudinal zonation.

## 1 Introduction

Accurate estimation of snow cover, being a phenomenon at the interface between the atmosphere and the land surface, is of particular interest to meteorologists and hydrologists. From a meteorological point of view, as soon as snow cover occurs, it

alters, e.g., surface radiation budget and roughness, which in turn affect other meteorological elements such as near-surface air temperature or wind speed. From a climatological perspective, seasonal variation of snow cover duration and snow depth are used as one of the climatological characteristics of winter. Although in numerical modelling it is usually snow water equivalent (SWE) which is a principal variable storing information about the amount of snow at every model timestep (ECMWF, 2016b; Le Moigne et al., 2018), there are issues in favour of considering snow depth instead. First of all, snow depth in situ measurements are taken more easily than SWE – beside a snow stick, no special equipment is needed. Secondly, they are more temporally continuous since there is no minimum amount of snow, below which taking a measurement is unfeasible. Consequently, the in situ data coverage for snow depth is higher than in case of SWE. Last but not least, it is snow depth which is actually assimilated in most of the models, as the model background field is transformed from SWE to snow depth prior to performing snow analysis (ECMWF, 2016a; Helmert et al., 2018).

In recent years, there has been a tremendous growth of studies which use reanalyses as a proxy for observations. Hence, reanalyses validation becomes a fundamental task. There are, however, several factors which make comparing point snow depth measurements with gridded data sets more complicated than in case of other meteorological elements, such as 2 m air temperature or precipitation. First of all, spatial heterogeneity of snow depth is much larger than the size of a grid cell of most existing datasets. Hence, what is actually compared is a point measurement against a grid-averaged value (Copernicus Climate Change Service, 2019). Secondly, snow depth is a cumulative parameter. As a result, bias between modelled and measured snow depth, assuming no data assimilation, has cumulative nature. It makes verification a non-trivial task. Naturally, the problem occurs at most in areas with snow cover persisting throughout a winter season and is less severe where snow cover is intermittent.

Despite the above-mentioned hurdles, accuracy assessment of snow in ERA5 and ERA5-Land reanalysis using point measurements from meteorological stations has already been carried out in literature. A study conducted by the provider of the reanalysis showed that both reanalyses underestimates snow at mountain sites, however, at altitudes up to 1500 m a.s.l. ERA5 generally outperforms ERA5-Land (Muñoz-Sabater et al., 2021). For more elevated sites, ERA5-Land is more accurate due to the added value of higher resolution. It is also shown that the superiority of one reanalysis against the other is region-dependent. Overestimation of snow in ERA5 has been identified in several studies over the High Mountain Asia (HMA) region (Lei et al., 2023; Orsolini et al., 2019; Wang et al., 2021). Lei et al. (2023) stated that the bias is especially distinct during the ablation period. They also spotted it in ERA5-Land, however, to a much lesser degree than in ERA5. Smaller overestimation of snow in ERA5-Land was also recognized over the mountainous part of Iran by Majidi et al. (2025). On the other hand, they detected a systematic underestimation in the atmospheric ERA5 across all elevations. Opposite conclusions were drawn by Monteiro and Morin (2023) who examined snow bias in the Alps. They found that it is ERA5-Land that has greater positive bias. The bias increases with elevation, while in ERA5 it becomes negative at altitudes below 600 m a.s.l. Overestimation of snow in ERA5-Land was reported in the Alps also by Pflug et al. (n.d.) and in the Atlas Mountains in Africa by Baba et al. (2021). However, it should be noted that those two works concern only SWE. Particularly interesting is research done by Varga and Breuer (2023) as it focused on the Pannonian Basin, which is just to the south of the area of this study. They claimed that

ERA5-Land overestimates snow while ERA5 matches observations well. However, their analysis was limited only to flat terrain. In addition, they concluded that the quality of ERA5 away from assimilated sites is yet to be investigated.

Monumental intercomparisons concerning also other than ERA5 regional and global reanalyses have shown that systematic biases in snow estimation are a common issue (Monteiro and Morin, 2023; Mudryk et al., 2025). Hence, there have been a couple of methods developed to reduce them. Some studies address it by dynamical downscaling, using WRF or some other regional numerical model (Avanzi et al., 2023; Fontrodona-Bach et al., 2023; Yang et al., 2021). Wang (2021) used the Japanese 55-year Reanalysis (JRA-55) to correct biased snow depth in ERA5 over the HMA region using a linear scaling factor. Recently, machine learning algorithms have been recognised to be a robust tool to achieve this goal, particularly Long Short-Term Memory (LSTM) and Random Forests (RF) models. King (2020) used RF to reduce overestimation of SWE in a regional 1-km resolution reanalysis in Ontario, Canada. Two studies concerning application of LSTM to forecast SWE based on meteorological forcing over the western United States reached the conclusion that without nudging the results with observations, LSTM predictions suffer from bias that grows over time (Cui et al., 2023; Song et al., 2024). It seems, however, that application of ML is the most advantageous for prediction based on combining data from multiple datasets and sources. Qiao et al. (2021) merged data from ERA-Interim, MERRA-2, GLDAS and microwave remote sensing to obtain snow depth over China. A similar job was done in the Western Himalayas by Tanniru et al. (2023) as they ran an extremely randomised trees model combining data from ERA5, MERRA-2, CMC and JRA-55 reanalysis. A broader study area occurs in work of Hu et al. (2024) where they combined several reanalyses (e.g., Era-Interim, MERRA2) and remote sensing snow products to create a gridded dataset for the Northern Hemisphere using RF. The results were superior to input reanalyses, however, the resolution was quite coarse (0.25°). Work on the integration of satellite products and data from ERA5 using LSTM to get continuous spatial distribution of snow depth in the Atlas Mountains is ongoing in Morocco (Elyoussfi et al., 2023).

In summary, studies investigating snow depth and SWE in ERA5 and ERA5-Land revealed it is mostly overestimated, with ERA5-Land being generally biased more. However, as pointed out by Muñoz-Sabater (2021), the exact outcome may be different as it is region- and elevation-dependent. Moreover, the work of Varga and Breuer (2023), which is to our knowledge the only one concerning Central Europe, is limited only to flat terrain and synoptic stations. A limited amount of research in this region regard also ML-based bias correction of snow estimation in reanalysis. Most publications on this topic concern HMA or western USA. Although tools proposed there are usually robust as they use information from more datasets than in this study, the output resolution is often quite coarse, especially with respect to the spatial scale of topographic complexity in Central Europe. Therefore, two primary objectives of this study are established: 1. To assess spatial and temporal variability of bias in ERA5 and ERA5-Land snow depth estimation in Central Europe 2. To explore the potential of ML to combine information from these two reanalyses in order to reduce the identified errors and create a consistent snow distribution over a subdomain with complex orography.

This paper has been divided into four parts. The first part describes input data and how it is processed to become predictors for a ML training. Then, a RF algorithm as well as verification metrics are briefly introduced. Additionally, a setup for a spatial downscaling experiment is defined. The Results section deals with assessing the accuracy of both reanalyses and the RF model.

100

110

It is followed by the discussion which handles e.g. circumstances favouring large deviations and the influence of data assimilation. Eventually, limitations of emulating spatial variability of snow in high resolution in complex terrain are carefully analysed.

#### 2 Data and methods

## 2.1 Study area

The investigated area extends from 48.5 to 55°N and 14 to 24.5°E, representing the north-eastern part of Central Europe. It covers the whole administrative area of Poland and most of the Czech Republic and Slovakia. Its northern and central parts belong to the North European Plain and are usually flat or gently rolling with elevations not exceeding 200 m a.s.l. (Fig. 1). South of that, uplands and mountain ranges occur belonging to the Central European Uplands. Two major mountain ranges can be distinguished: the Sudetes and the Western Carpathians. Their elevation ranges from 800 to 1600 m a.s.l. in the Sudetes and from 800 to 2650 m a.s.l. in the Western Carpathians. The south-western edge of the study area reaches also the Bohemian Forest (800 - 1400 m a.s.l.). Average snow conditions in the study area are characterised in the Appendix A.

Fig. 1. Orography map of the study area with country borders and stations included in the study (S – synop, C – climate, P – precipitation)

## 2.2 Study period and ground-based data

The study spans 20 winter seasons defined as a period between 01.11 and 30.04 in the years 2001-2021. While the assumed duration of a winter period is certainly too short for high-elevated sites (particularly regarding a snowmelt period), it was set

© Author(s) 2025. CC BY 4.0 License.

as a reasonable compromise between losing some data and a large amount of empty data (i.e., without snow) from the majority of the stations. The initial and the final year of the study period were specified so as to capture considerable interseasonal winter variability while having limited access to historical snow depth measurements in national weather centres. A more detailed description of snow conditions in the study period may be found in the Appendix A.

Ground-based data includes snow depth measurements taken daily at 6 UTC at meteorological stations of national weather services of Poland, Czech Republic and Slovakia. Only data from manned stations were collected. The stations are grouped according to their importance into three ranks: synoptic (reporting every hour according to an international SYNOP standard), climate (reporting several basic meteorological elements internally at 6,12 and 18 UTC) and precipitation (reporting only daily precipitation sum and snow phenomena internally at 6 UTC). The methodology of snow depth measurements is the same across the station ranks, however, due to diverse quality control mechanisms, credibility of data is somewhat lower for non-synops. Additionally, measurements at a few top-mountain stations are done in an approximate way due to site's high susceptibility on wind-induced erosion. A total of 340 stations (91 synoptic, 140 climatological and 109 precipitation) were selected according to the following criteria (ranked from most to least important):

- average snow cover duration
- data completeness throughout the study period
- relatively even spatial distribution of stations in the study area
- · station rank

Consequently, stations at mountainous sites were included even if they worked only for a few winter seasons. For this reason, a denser station network could be seen in the southern part of the study area (Fig. 1). If two stations with complete data records but different ranks were close to each other, the higher-ranked one was picked. Although station elevation ranges in total from 1 to 2635 m a.s.l., 50% of the stations lie below 300 m a.s.l. and only 5% above 1000 m a.s.l.

## 2.3 ERA5 and ERA5-Land Reanalyses

A number of meteorological and topographic fields from ERA5 and ERA5-Land reanalysis have been retrieved using the Climate Data Store API. Both datasets belong to the ERA5 family, which is the latest (as of the time of submitting the manuscript) generation of global reanalysis of past weather conditions produced by ECMWF (Hersbach et al., 2020). A full list of parameters from both reanalyses is to be found in Table B1 (Appendix B).

ERA5 is an atmospheric reanalysis combining short-ranged numerical forecasts with meteorological observations during a process of data assimilation (DA). It has 0.25° horizontal resolution (around 30 km) and 1 h temporal resolution. It assimilates snow depth observations from synop stations and some additional national snow measurements that are available on the Global Telecommunication System (GTS) (ECMWF, 2016a; de Rosnay et al., 2015). As claimed by the ECMWF Support Portal (2024), no other measurements than synops are assimilated from the area of Poland, Czech Republic and Slovakia. The observations are then combined with a snow-depth background field (estimated from forecasted SWE and snow density) and

5

a satellite-based snow extent product (ECMWF, 2016a). The Optimal Interpolation algorithm is used for that purpose. The analysis is performed every 6 h starting at 00 UTC. It is worth noting that the snow extent product was not included in the DA system for data before 2004, which is a potential source of discontinuity. Furthermore, for data after 2010 its horizontal resolution was upgraded from 24 to 4 km (Mortimer et al., 2020; de Rosnay et al., 2015). Additionally, before the exact DA, observations undergo various quality checks. The most important ones regarding snow are that the maximum threshold of 140 cm for in situ snow depth data is given (it is lowered to 70 cm if a background 2-m air temperature exceeds 8°C) and the satellite-based snow extent is not assimilated from areas above 1500 m a.s.1 (ECMWF, 2016a). In the output dataset, snow depth occurs in fact as SWE – it is defined as the depth of water after the whole snow melted and was uniformly spread over a grid box. To obtain snow depth sensu stricto, it is necessary to divide the field by snow density. Hence, bias in snow density may be transmitted to snow depth.

ERA5-Land is a land surface reanalysis run in an offline mode with an atmospheric forcing coming from the ERA5 dataset (Muñoz-Sabater et al., 2021). Output fields are produced with a 1-hour frequency at the enhanced 0.1° horizontal resolution, which is roughly three times higher than in case of ERA5. No DA is deployed in ERA5-Land. Observations are indirectly passed to the model through an atmospheric forcing. The forcing includes surface pressure, air temperature, specific humidity, wind components, liquid and solid precipitation as well as solar radiation fluxes. It is important to highlight that no snow fields are passed to ERA5-Land. Just like in ERA5, actual snow depth is retrieved from two prognostic variables: SWE and snow density.

## 165 2.4 Auxiliary data

Data from the Digital Elevation Model (DEM) - Shuttle Radar Topography Mission Version 4.1 were used to provide information about altitude and topography (Jarvis et al., 2008). The data is in 3" horizontal resolution (around 75 m in the study area).

## 2.4 Data preprocessing

## 170 **2.4.1 Ground-based data**

If a trace of snow (i.e., snow depth lower than 0.5 cm) or snow in patches (i.e., less than one-half of the ground covered by snow) was reported, snow depth was assumed to be 0 cm. If snow cover was discontinuous but it covered more than one-half of the ground, snow depth was kept as measured.

## 2.4.2 ERA5 and ERA5-Land reanalyses

Reanalysis data were extracted in the nearest grid point for every station using the Climate Data Operators (CDO) function *remapnn* (Schulzweida, 2023). Next, selected fields were temporally aggregated in order to capture their variability in between snow depth measurements, which are taken once a day at 6:00 UTC. The exact metric of aggregation and its timeframe have

been subjectively defined by the authors depending on the nature of a meteorological element and its potential impact on snow depth. See Table B1 (Appendix B) to find out how exactly every reanalysis parameter has been aggregated.

O Since the nearest grid point for a few coastline stations is covered predominantly by a sea, a number of land variables are not available there. Therefore, an automatic search of the nearest land grid point was performed and assigned to a station. Most often it was one of the other surrounding grid points. One station have to be removed from the database as it lies at the tip of a 35-km-long sandspit that expanded into the Baltic Sea and therefore, a shift by 0.4° would be required.

## 2.4.3 Auxiliary data

The following topographic parameters were calculated based on the DEM: topographic position index (TPI), sky view factor (SVF) and a fraction of the surrounding area with higher elevation. Dedicated R packages *terra* and *horizon* were used for that purpose (Hijmans, 2020; Van Doninck, 2018). Computations were performed for every grid point for a set of user-defined radii ranging from 100 to 30000 m in order to capture land relief variability at different scales. However, due to high computational cost, the calculation of SVF and a fraction of the surrounding area with higher elevation was limited to 5000 m.

Data from DEM were also used to calculate the daily-integrated amount and duration of direct incoming solar radiation. This was performed using a Points/Area Solar Radiation algorithm available in the ArcGIS Spatial Analyst Toolbox (ArcGIS 10.2.1). Apart from DEM-based predictors, a few temporal and station-embedded variables were also included in the dataset in order to emulate the passing of time and spatial distribution of data, respectively. The exact list of parameters and their aggregations is to be found at Table B2 (Appendix B).

## 195 2.5 Machine learning

The RF algorithm designed by Breiman (2001) and implemented in *R* environment (package *ranger*) by Wright and Ziegler (2017) is used in the study. It generates an ensemble of decision trees by bootstrapping the training data, which are then aggregated into a final prediction. Generalisation skill and performance accuracy are estimated on a training set using an out-of-bag error (Breiman, 2001). Along with neural networks, RF has been commonly deployed at various stages of weather and climate modelling (Bochenek and Ustrnul, 2022; de Burgh-Day and Leeuwenburg, 2023). In this study, we use it to predict an absolute value of snow depth, which is a regression problem (Fig. 2). The initial set of around 120 predictors was prepared based on meteorological knowledge of physical processes affecting snow depth. It was eventually shrunk to 70 predictors in a process of feature selection to account for redundancy and multicollinearity. The selection was made based on the Boruta algorithm implemented in the *R* package *Boruta*, permutation variable importance from the *ranger* package and Variance Inflation Factor implemented in the *collinear* package (Benito, 2023; Kursa and Rudnicki, 2010). The predictors are listed in Table B1 and B2 (Appendix B). Next, an automated tuning of training hyperparameters from the *tuneRanger* package was performed to find the best model settings (Probst, 2025). The tunable parameters are: minimum node size (*min.node.size*), the number of predictors at each split (*mtry*) and the fraction of observations at each split (*sample.fraction*). The number of trees in the forest was set to 1000. A set of parameters that gives the lowest out-of-bag error was picked as the most optimal

configuration, which is mtry=31, min.node.size=2 and sample.fraction=0.895. Predictions of the RF model for the whole dataset (20 winter seasons) were produced using a leave-one-year-out cross-validation. This is a particular example of k-folds cross-validation method that is often applied when handling time-series meteorological data (Hu et al., 2024; Sebbar et al., 2023). A set of 20 RF models were generated where each winter season was treated as a test set, while the remaining 19 seasons were used for training. No spatial split was applied – the stations in the test set are the same as in the training set.

Fig. 2. A schematic diagram of data used for training of a RF model. Only station data was used to train the model.

## 2.6 Spatial downscaling experiment setup

Apart of point verification on station locations, the above described RF model was tested also over a small subdomain in complex terrain to assess its skill in downscaling of ERA5 snow depth estimations. The domain covers the Tatra Mountains – the highest mountain range in the Carpathian Mountains located on the border between Poland and Slovakia (Fig. 3). Several factors determined this selection. First of all, due to its elevation, it is the largest snow reservoir in the study area. Hence, accurate estimation of snow amount is crucial there. Secondly, the horizontal resolution of both reanalyses is much lower than the scale of orographic complexity of most of the area, which results in large biases that need to be reduced. Hence, it was expected that the added value of the DEM would benefit here the most. Furthermore, there is a considerable spatial coverage of observations in this area which makes accuracy evaluation more reliable.

Fig. 3. A subdomain selected for a spatial experiment (Tatra Mountains). Stations are coloured according to their rank (S - synop, C - climate, P - precipitation)

The predictions of RF for the experiment were generated using a model trained on stations (as shown in Fig. 2). Thus, the final outcome is in fact just an ensemble of point predictions produced separately for every grid point of the domain. Prior to it, all predictors in a test set were bicubicly interpolated to the DEM grid (3") using CDO software (Schulzweida, 2023). The predictions are then analysed only at one timestep, primarily for demonstrating potential spatial skills rather than thoroughly assess them, which could be a subject of separate research. Additionally, in light of obtained results, sensitivity analysis of wind-related predictors (daily mean wind speed, wind direction and daily maximum wind gusts at 10m) was conducted.

## 235 2.7 Evaluation metrics

To assess the accuracy of the predictions and the reanalyses, commonly known verification metrics such as Root Mean Square Error (RMSE), Mean Absolute Error (MAE), Mean Error (bias), standard deviation of bias (SD) and MAE Skill Score (SS<sub>MAE</sub>) are used in the study (Table 1). RMSE is by far the most commonly used benchmark metric employed to evaluate reanalysis or forecast accuracy, however, as it penalises greater deviations more, errors occurring at mountainous stations would affect the global metric more than in case of MAE. Hence, it was decided to use MAE as a basic verification metric in the study. Another snow-specific issue caused by the cumulative nature of snow is an increase of error with an increasing amount of snow (provided no DA is applied). To account for that, normalisation is carried out (Monteiro and Morin, 2023; Muñoz-Sabater et al., 2021). In the study, mean snow depth is used to normalise MAE (denoted as nMAE). To avoid the influence of snowless days, all metrics were calculated only for data when either measured or reanalysed snow depth was above 0 cm.

Table 1. Verification scores used in the study (adapted from Warner (2011)). Denotations: N - number of data samples, F<sub>i</sub> - i-th forecasted value, O<sub>i</sub> - i-th observed value, MAE<sub>RF</sub> - MAE of an evaluated RF forecast, MAE<sub>ref</sub> - MAE of the best reanalysis.

| Metric                | Formula                                                                                                                                   |  |  |
|-----------------------|-------------------------------------------------------------------------------------------------------------------------------------------|--|--|
| MAE [cm]              | $\frac{1}{N}\sum_{i=1}^{N} F_i-O_i $                                                                                                      |  |  |
| bias [cm]             | $\frac{1}{N}\sum_{i=1}^{N}(F_i-O_i)$                                                                                                      |  |  |
| RMSE [cm]             | $\sqrt{\frac{1}{N}\sum_{i=1}^{N}(F_i-O_i)^2}$                                                                                             |  |  |
| SD [cm]               | $\sqrt{\frac{1}{N}\sum_{i=1}^{N}(F_i-O_i-bias)^2}$                                                                                        |  |  |
| nMAE[cm]              | $\frac{MAE}{\frac{1}{N}\sum_{i=1}^{N}O_{i}}$                                                                                              |  |  |
| SS <sub>MAE</sub> [%] | $\frac{\mathit{MAE}_{\mathit{RF}} - \mathit{MAE}_{\mathit{ref}}}{\mathit{MAE}_{\mathit{perf}} - \mathit{MAE}_{\mathit{ref}}} \cdot 100\%$ |  |  |

## 3 Results

Considering overall verification scores from Table 2 (bottom row), estimation of snow depth is more accurate in ERA5-Land than in ERA5. While the difference in MAE is relatively small, it is much higher in case of bias. To address this issue, metrics in relation to elevation need to be examined. Aggregated metrics for elevation intervals (Tab. 2) as well as plots for bias and MAE (Fig.4a-d) show a clear increase of error with elevation. Particularly for elevations over 1000 m a.s.l., the trend is roughly linear. Both reanalyses severely underestimate snow depth there, however, the deviations are slightly lower in ERA5-Land. What differentiates the datasets the most is bias tendency below 1000 m a.s.l. While in ERA5-Land it is predominantly positive (only 15% of stations experiencing underestimation of snow), in ERA5 it is diverse with substantial negative bias at some sites at 500-1000 m a.s.l. Regarding MAE, it is interesting to notice that although the global score from Tab. 2 gives favour to ERA5-Land, at 65% of stations it is actually ERA5 that performs better. The ratio increases even to 75% if stations below 500 m a.s.l. are considered. In zoomed areas in Fig. 4c-d, stations are coloured according to their rank. A couple of issues are striking here. Errors in ERA5 can go as low as around 0.5 cm, whereas in ERA5-Land it is always higher than 2 cm.

Additionally, the majority of the lowest errors in ERA5 occur at synop stations (red dots), while in ERA5-Land no clear separation is visible. This could be directly attributed to data assimilation. Thirdly, for ERA5 there is an error spike near the sea level, which is barely distinguishable for the other reanalysis. The rise of error affects also synops, which could be explained by the fact that these stations were dismissed from data assimilation due to their location at the interface between land and sea.

Table 2. Average accuracy of snow depth estimation in ERA5 and ERA5-Land.

| elevation<br>interval | number of stations | database         | MAE<br>[cm] | RMSE<br>[cm] | bias [cm] | SD [cm] |
|-----------------------|--------------------|------------------|-------------|--------------|-----------|---------|
| < 250                 | < 250 146          | ERA5             | 2.65        | 4.23         | 0.25      | 3.67    |
| < 230                 |                    | ERA5-Land        | 3.86        | 5.77         | 2.33      | 5.28    |
| 250 500               | 250 500 02         | ERA5             | 5.52        | 9.2          | -0.27     | 8.06    |
| 250-500 92            | 92                 | ERA5-Land        | 7.47        | 11.7         | 6.16      | 9.95    |
| 500-1000              | 500 1000 00        | ERA5             | 12.52       | 20.25        | -6.82     | 17.53   |
| 300-1000              | 88                 | ERA5-Land        | 11.76       | 18.21        | 3.71      | 17.82   |
| 1000-                 | 1000-<br>1500 8    | ERA5             | 36.99       | 52.69        | -36.31    | 37.97   |
| 1500                  |                    | ERA5-Land        | 30.83       | 45.81        | -25.29    | 38.2    |
| > 1500                | (                  | ERA5             | 85.7        | 109.37       | -85.56    | 68.4    |
|                       | 6                  | ERA5-Land        | 69.76       | 92.98        | -69.09    | 62.24   |
| overall               | 340                | ERA5             | 11.43       | 27.44        | -7.13     | 23.88   |
|                       |                    | <b>ERA5-Land</b> | 10.5        | 22.56        | 0.62      | 22.55   |

To compare MAE between stations in a more objective way, the metric was normalised with the mean snow depth for each station (Fig. 4e-f). As a result, the relationship with elevation is more complex, and no clear trend is visible anymore, especially for ERA5-Land. In both datasets, the highest errors occur at an altitude range of 400-700 m a.s.l. These are predominantly stations lying in the foothills of the Carpathian Mountains and the Sudetes rather than high mountainous ones. If a threshold value of nMAE=1 would be considered, there are only three sites exceeding it in ERA5: two lower-rank ones in the Sudetes (Karpacz, Przesieka) and one Slovakian synop Poprad-Gánovce (Fig. 5). For ERA5-Land the threshold is violated by a number of stations in mountain valleys or basins, including synops, e.g., Zakopane, Liesek or Kłodzko. The error for these sites in ERA5 is usually 3-4 times lower likely owing to data assimilation.

Fig. 4. Bias (top row), MAE (middle row) and nMAE (bottom row) at every station in the database in relation to elevation for ERA5 (left column) and ERA5-Land (right column). Stations at which a given reanalysis performs better than the other are shown in green, while those where it performs worse – in grey. In zoomed areas in the middle row, stations are coloured according to their rank (S - synop, C - climate, P - precipitation).

Fig. 5. Location of stations with nMAE of snow depth estimation exceeding 1 cm. Stations are coloured according to their rank (S - 280 synop, C - climate, P - precipitation). The colour of the label denotes reanalysis, for which the error occured (blue - ERA5, light green - ERA5-Land, purple - both).

Having analysed the elevation-dependent differences in both reanalyses, now the interseasonal variability will be investigated (Fig. 6). Seasonal MAE averaged across all the considered stations changes substantially throughout the seasons. Both reanalyses correlate well with each other with respect to this metric. The highest values occur in the seasons 2004/2005, 285 2005/2006, 2011/2012 and 2018/2019, which were all snowy ones. The lowest MAE refers to the 2013/2014 season, which was in fact one of the least snowy ones, however, also the 2009/2010 season, which was actually quite abundant in snow. Additionally, the season 2019/2020, which is by far the least snowy, has relatively large errors, particularly in ERA5. The error for RF predictions is lower by around 50% than for the reanalyses and exhibits similar changes in relation to the mean snow depth. To remove this impact, nMAE was also analysed (Fig. 6b). After normalisation, the intraseasonal fluctuations are less severe and the errors are relatively steady in time. The lowest error for both reanalyses is achieved in the 2009/2010 season (for ERA5-Land also the 2004/2005 season). What is striking is a sharp drop of the error for the reanalyses from 2008/2009 to 2009/2010. As regards RF, two seasons are easily distinguishable by an increased value of the error: 2010/2011 and 2013/2014. Since the reanalyses performed quite well then, the relative improvement (SS<sub>MAE</sub>) is much smaller than the average (11% and 31%, respectively). This could be attributed to the unusual winter conditions during that time. However, this is barely visible on boxplots as the conditions occurred mainly at mountainous stations, which are scarcely represented yet provide a lot of data with snow. The two seasons turned out to be the driest ones in the study period and involved frequent melting episodes (Fig. 7). Consequently, the evolution and accumulation of snow cover was significantly deviated then, particularly regarding the second part of the seasons.

In total, RF estimated snow depth with the mean bias of 0.26 cm and MAE - 5.5 cm, which provides a large improvement upon both reanalysis (compare with the scores in Table 2). Metrics calculated for each station show that RF is superior at 70%

of them, while at the majority of the remaining 30%, RF is outperformed by ERA5 only slightly (less than 0.5 cm of MAE difference). These are mostly synops or lower-rank stations strongly affected by DA and thus having very small errors and therefore little space for improvement.

Fig. 6. Seasonal variability of MAE (a) and nMAE (b) for ERA5, ERA5-Land and RF predictions (lines),  $SS_{MAE}$  of RF over the best reanalysis (barplot) and the mean snow depth for all stations (boxplot).

Fig. 7. Temporal evolution of snow depth in seasons 2001/2002 - 2020/2021 for Štrbské Pleso, Kasprowy Wierch and Śnieżka synop stations. The two bolded lines represent 2010/2011 and 2013/2014 seasons.

As far as the downscaling experiment is considered, the fine-scaled spatial distribution of snow depth on 6 December 2016 at 6:00 UTC produced by RF based on coarse-resolved reanalysis ERA5 and ERA5-Land is depicted in Fig. 8b. For comparative purposes, snow depth from ERA5-Land with native grid mesh and land relief is also shown (Fig. 8a and c, respectively). The date was chosen based on observations availability and climatological characteristics of the season. Moreover, an early stage of snow accumulation was deliberately selected not to violate a 140 cm threshold of maximum snow depth allowed in analysis during the DA process in ERA5 (ECMWF, 2016a). The predictions were initially supposed to be verified with spatially interpolated measurements. However, probably due to high terrain complexity, none of the interpolation methods yielded

https://doi.org/10.5194/egusphere-2025-5084 Preprint. Discussion started: 24 November 2025 © Author(s) 2025. CC BY 4.0 License.

meaningful results. Consequently, verification is limited only to station points. The forecasted and measured values of snow depth are shown on the map in red and black, respectively. What is easily noticeable is the elevation dependence of the field. The greatest spatial variability occurs in highly complex terrain. In the northern part of the domain, where elevation does not vary that much, the predictions are quite close (14-29 cm). However, the true variability in this area, evidenced by the station snow depth values, is larger. This is particularly noticeable at two neighbouring sites, Poronin and Bukowina Tatrzańska. Despite only 4 km of horizontal distance and around 130 m of elevation change, snow depth in Bukowina Tatrzańska is 18 cm thicker than in Poronin. This value was considerably underestimated in RF predictions. Similar bias could be observed for Białka Tatrzańska, which is just north. Interestingly, opposite bias occurs at Bańska Wyżna. Despite considerable elevation (896 m a.s.l.), there is less snow there than at lower-elevated stations in the vicinity. Such a deviation from the elevationdependence relation is likely to be a source of RF misprediction. As far as stations in the Tatra Mountains are considered, deviations slightly increase, ranging from a few up to 21 cm. There is one site with an exceptionally large discrepancy between a prediction and the true value - Lomnický Štít. This is a mountain-top station at the highest altitude in the dataset (2635 m a.s.l.). Snow depth is severely underestimated there. Apparently, the RF model prodicts the Eastern Tatras to have slightly less snow than the western part, which stands in contrast to observations as well as typical snow conditions there. This might imply a systematic error that may be derived from the reanalysis (as the SE corner of the domain had the least snow in ERA5-Land).

Closer inspection of Fig. 8b shows another interesting issue – the RF model accumulated more snow on northern slopes. Hence, it might be concluded that it was able to emulate the shadowing effect of the orography. This is to be seen at best along eastwest-oriented ridges, particularly along the main ridge of the Western Tatras, which is shown in magnification in Fig. 9. This is likely to be an effect of including DEM-based parameters like potential solar radiation and SVF into the training dataset.

Fig. 8. Spatial variability of snow depth on 6th December 2016 6:00 UTC in ERA5-Land (a) and the result of downscaling by RF (b). Measured and predicted values for stations are shown in black and red, respectively. The bottom map depicts land relief (for comparison purposes only). Stations are coloured according to their rank (S - synop, C - climate, P - precipitation).

Fig. 9. Spatial variability of snow depth along the main ridge of the Western Tatras (a) and its orography (b). The main valleys are marked with numbers: 1 – Chocholowska Valley, 2 – Kościeliska Valley, 3 – Tichá Valley.

Apart of the shadowing effect, it is commonly acknowledged that above the tree line, it is wind redistribution that affects snow variability the most. However, wind-related predictors were excluded from the RF training dataset due to its poor importance score. The 25 most informative variables are ranked in Fig 10. Except of snow depth itself, topographic variables that carries information about elevation are most precious. What is worth noticing is high values of parameters cumulated from the season beginning (snowfall, snowmelt, total precipitation). Information from some past time is also included in other high-ranked predictors such as age of the snow depth, mean air temperature in the second soil layer and snow density. Wind-related predictors were scored way below 1. Therefore, a sensitivity analysis for these predictors have been additionally conducted.

The predictors were added to the training dataset listed in the Appendix B and a new RF model was trained. Then it was compared in selected periods where weather conditions favored wind redistribution. The conditions were: average daily wind speed over 1.5 m/s, maximum wind gusts over 10 m/s, daily maximum temperature below 0°C, daily precipitation below 0.5 mm and observed daily snow depth reduction by at least 5 cm. In total, 40 such cases were identified. The verification was performed for one station Hala Gasienicowa, since this station lies above tree line and its snow depth report is representative (averaged from five snow poles). The place is also known for snow being blown away to lower parts of the valley which results in snow depth reduction. The mean snow deflation as seen by RF is 2.1 cm while in fact it was over 11.5 cm. This is only slightly better than the reference RF model (with no wind-derived predictors), which reduced snow depth by 1.6 cm on average. ERA5-Land does not exhibit any snow depth reduction (0 cm on average). Global metrics calculated for the stations within the subdomain show minor improvement in most cases (differences in MAE predominantly below 0.1 cm).

Fig. 10. Importance score for the 20 most important predictors of the reference RF model. Predictors were manually grouped into categories denoted with colors. Variable abbreviations are explained in Table B1 and B2. Following suffixes are used: cum – cumulative parameter, age – snow age, avg24 – daily average value. The X-axis is logarithmic.

#### 365 4 Discussion

#### 4.1 Bias in mountainous terrain

Underestimation of snow depth at high elevations (above 1000 m a.s.l.) in both ERA5 and ERA5-Land has been reported, e.g., by Muñoz Sabater et al. (2021) and was attributed to orography smoothing. As it is an effect of orography discretisation, it occurs in most numerical models (depending chiefly on their horizontal resolution). In this light, less deviation observed in 370 this study at high elevations in ERA5-Land could be clearly explained by better resolution of this reanalysis, which is in line with conclusions of Muňoz Sabater et al. (2021) or Lei et al. (2023). However, their evaluation was done either globally or for a specific mountain area located remotely from the study area. Surprisingly, conclusions from studies regarding the Alps, which are the closest large-scale mountain range, stay in contradiction to our findings (Dalla Torre et al., 2024; Monteiro and Morin, 2023; Shrestha et al., 2023). This clearly demonstrates that the impact of orography smoothing could be completely offset by some other factors, which can locally become a dominant source of deviation. Bias in snow depth may be derived from various factors, out of which the most important are generally errors of air temperature, precipitation or shortcomings in parametrisation of snowpack properties, particularly snow density. The last one is usually systematic (spatially invariant), which does not explain the current discrepancy. Hence, the reason might be either temperature or precipitation bias that differs between the Alps and the Carpathian Mountains. As a matter of fact, in comprehensive pieces of work by Dalla Torre et al. (2024) and Monteiro and Morin (2023) these two elements were also evaluated. The authors identified substantial wet bias that contributes to overestimation of snow depth the most. Lack of accuracy assessment of air temperature and precipitation in this study hinders a clear explanation of the observed bias, which is a major limitation of the work.

#### 4.2 Role of data assimilation

Outside complex terrain, the differences in errors between ERA5 and ERA5-Land are determined chiefly by the presence of DA in the coarser-resolved reanalysis. It is important to notice that despite the fact that only synop stations are assimilated, the improvement is visible also for lower-rank stations. First of all, it should be reminded that apart of SYNOP reports, additional information about snow extent is provided by satellites. However, this is just binary information with no plain conversion to snow depth (de Rosnay et al., 2015). From the IFS documentation (2016a), one can learn that horizontal and vertical structure functions applied in the Optimal Interpolation algorithm play an essential role in interpolating snow depth values in space between assimilated points. Because of them, the influence of a point measurement assimilated in the model spreads to neighbouring areas. It was examined more deeply by Stachura (2024), who claims that it yields satisfactory results unless there is a considerable elevation difference between a low-ranked station and an assimilated synop station.

However, some synops are excluded from the assimilation process. Our findings suggest that this is the case for sites at the sea

coast. More importantly, there are several quality control conditions which result in rejection of measurements from, e.g., mountainous stations (ECMWF, 2016a). The IFS documentation does not specify a certain elevation threshold, below which measurements are accepted. Based on MAE scores presented in the study, one can state that the impact of DA is to be seen

https://doi.org/10.5194/egusphere-2025-5084 Preprint. Discussion started: 24 November 2025 © Author(s) 2025. CC BY 4.0 License.

up to around 900 m a.s.l., however, our analysis of the normalised MAE revealed that there exist synops lying below this altitude with errors suggesting they were most likely not included in the DA system (e.g., Poprad-Gánovce). This indicates that quality control of snow depth DA in ERA5 involves thresholds other than elevation-based one.

## 400 4.3 Random Forests performance

Analysis of interseasonal variability of nMAE and SS<sub>MAE</sub> indicated two seasons with relatively weaker performance of RF with respect to the reanalysis. This deviation was attributed to very dry conditions in the southern part of the study area during these two winters. Consequently, the process of snow cover accumulation in the mountains was abnormal. As a matter of fact, some sites recorded the seasonal maximum just at the beginning of December. Since RF predictions were generated through a leave-one-year-out cross-validation, for each of the two abnormal seasons, one of them was always included in the training set. Hence, it was not the case that such circumstances were completely unknown for the RF model. Apparently, however, they were too rare or too local for the model to generalise the information and represent it correctly. The importance of an equally balanced training set is commonly acknowledged in machine learning (Meehan et al., 2024; O'Gorman and Dwyer, 2018). One possible solution could be to use oversampling techniques that make training sets well-balanced. Otherwise, some other ML method could be deployed as some studies suggest that ANN or DL may be more robust in similar circumstances (Pflug et al., n.d.; Stachura et al., 2024).

Nevertheless, considerably lower  $SS_{MAE}$  for the two specific seasons is not only an effect of RF mispredictions but also exceptionally low bias of the reanalysis. As concluded from Fig. 4, snow cover in mountainous terrain is generally heavily underestimated. From the other site, dry winters result in poor amounts of accumulated snow during the season. Bearing in mind the cumulative nature of snow, this eventually leads to lower bias of the reanalyses.

#### 4.4 Spatial downscaling experiment

Station verification shown in Fig. 8b is in such complex terrain definitely insufficient to draw a conclusion that spatial distribution of the downscaled snow depth is legitimate and corresponds to the ground truth in the whole subdomain. It was shown that except for elevation dependence, also the shadowing effect is to be noticed due to including predictors which take into account the topographic influence of the surrounding terrain. This corresponds well with variable importance presented in Fig. 10 where information about elevation and solar radiation belong to the most informative predictors. However, both variable importance and the sensitivity test proved little or negligible impact of the wind, while it is commonly acknowledged that this is a predominant determinant of spatial variability of snow, especially above the tree line (Mott et al., 2018). First of all,, observation network above tree line is really scarce and hardly reflects wind redistribution in an objective way. As mentioned previously, some of the mountain-top stations do not measure snow depth directly (i.e., by reading the value from a snow pole) but rather estimate it based on fresh snowfall and other meteorological quantities so that the final value is representative for a greater area than a mountain peak itself. Such methodology is used, e.g., at Kasprowy Wierch, Śnieżka and Lomnický Štít. This, however, is misleading for a ML algorithm during training since the target values for these stations

https://doi.org/10.5194/egusphere-2025-5084 Preprint. Discussion started: 24 November 2025 © Author(s) 2025. CC BY 4.0 License.

have slightly different meanings than for stations which measure snow depth explicitly. One way to get around it would be simply to remove the problematic sites, however, it would result in significant shrinkage of the data, especially above 1500 m a.s.l. and with large values of snow depth. In view of the known limited capability of RF to predict extremes (Muckley et al., 2022; Sun et al., 2024; Yang et al., 2020), this may lead to the deterioration of the final results. Another solution would be to include measurement data from ultrasound sensors, which have been recently set up at several stations in Poland. Lack of long observation series could be partially offset by higher data frequency. Secondly, the majority of training data come from stations 435 located in lowland areas, where strong wind events are much less frequent than in mountainous regions and thus, wind redeposition occurs only rarely. A potential solution for this would be to train a ML model only on data from mountainous sites. This would result, however, in reduction of generalization ability of the model. Thirdly, as certain meteorological variables have proved to be more informative when expressed in their cumulative form, we speculate that a similar approach would be beneficial for wind-related predictors. However, aggregating them to this form would be more challenging than, e.g., precipitation, since effects of wind redistribution (deposition/erosion) are often wind-direction-dependent. A parameter accounting for these processed was developed and successfully used by Liu et al. (2025). However, it should be admitted that studies on snow wind redistribution in complex terrain using ML remain challenging and, therefore, relatively rare. More frequently, the topic is addressed through numerical modelling (Liston et al., 2007; Marsh et al., 2024). Apart from wind, there are a few other snow depth determinants which are completely unrepresented in observations due to adhering to the WMO standards for weather stations siting. These are, e.g., influence of slope steepness, snow avalanches or canopy. Consequently, the RF model has no information about how snow accumulates on steep slopes or in the forest. As a result, RF predicts considerable amounts of snow even on extremely steep slopes, which is obviously erroneous. Owing to a lack of target observations, the potential of a ML approach seems to be limited. However, some of these processes (e.g., snow interception over canopy or snow settlement over steep slopes) are parameterised in surface numerical models, e.g., SURFEX (Hedstrom and Pomeroy, 1998). Hence, it might be worth considering developing a hybrid approach which would combine the strengths of both ML and classic Numerical Weather Prediction (NWP) models, e.g., using physics-informed ML (Viallon-

## 5 Conclusions

Galinier et al., 2023).

In both reanalysis (i.e., ERA5 and ERA5-Land), heavy underestimation of snow depth is observed in complex terrain, where accurate estimation of snow is particularly crucial, as this is a key winter water reservoir for this part of Central Europe. On average, ERA5-Land is less biased than the atmospheric ERA5 due to to its higher resolution and the fact that mountainous stations are excluded from the DA process in the atmospheric reanalysis. In flatter terrain, the accuracy of snow depth estimation is predominantly affected by DA, which makes the coarse-resolved reanalysis better at 75% of sites below 500 m a.s.l. This highlights the importance of DA and indicates major deficiencies in parametrisation of snow processes. MAE for

both reanalysis generally increases with elevation, however, after normalisation the greatest errors occur at sites located at the foothill zone rather than high in the mountains.

Continued efforts are needed at the side of reanalysis providers to enhance the DA system in several aspects. First of all, at the level of international data collection policies, an effort is required to adapt them so that more national databases of snow depth (which also involve measurements from lower-ranked stations) would be available to be assimilated. Secondly, at the scientific level, more robust methods should be deployed to handle assimilating of measurements in areas which were so far masked out from the procedure (due to strict quality control requirements or other assumptions). It is important to note here that DA would not be that crucial to reanalysis accuracy if physical parameterisations of snow processes were not flawed. Moreover, it is commonly known in the NWP community that correcting for bad physics is not the primary goal of DA. Therefore,

improvement in the description of s.now-related physical processes is a challenge of the utmost importance.

On average, the RF model was able to reduce systematic errors occurring at both reanalyses by as much as 48%. The greatest improvement occur in elevated terrain, predominantly due to improved resolution of orography. In lowlands, the differences between model and real elevation are little and the observed improvement occur mostly due to simplifications in parametrizations of snow processes, particularly in non-assimilated areas. Stations where RF predictions were not superior are the stations with already small error, most likely due to the impact of DA. Importantly, RF diminishes interseasonal variations of the error which were apparent for the reanalyses. The study identified two winter seasons with distinctly smaller relative improvement and argued that abnormal climatological conditions contributed to it. This points out the importance of a well-balanced training dataset and a potential limitation in application of this method of ML. Spatial downscaling performed by RF in areas with complex orography produced a fine-scaled spatial distribution of snow depth. Despite a decent overall performance verified pointly at 20 sites and the ability to capture more snow accumulated on northern slopes, predictions turned out to be spatially underdispersive in areas where other factors than elevation affect snow depth. Additionally, there are some factors which are crucial in determining the spatial extent of snow depth in mountainous terrain at such resolution, but they are barely reflected in observations (e.g., snow avalanches, drifting snow). Consequently, information about these determinants in training data is incomplete and deficient.

Overall, the main limitation of the RF model presented in this paper is that it requires ground station measurements and therefore, its skill and generalization capabilities depend chiefly on the measurements quality and representativeness. This allows the conclusion that other sources of measurements and observations should be deployed in order to address the problem more properly in the future, preferably ones which are spatially continuous. Such data are provided by satellite measurements, e.g. passive microwave remote sensing or laser altimetry, and was used in snow depth estimation by, e.g., (Liu et al., 2025; Takala et al., 2011). However, other problem arises when processing this type of data such as scarce temporal coverage or too coarse resolution.

Finally, it should be added the above-mentioned limitations apply to virtually every modelling approach, especially when considering such a diverse climate element as snow cover and its thickness. In light of the literature and our own experience with snow cover measurements, it should be noted that the value of measurement data is crucial not only for simple spatial

analyses using GIS spatialization methods but also for the application of ML methods, including RF. Nevertheless, we believe the experiment provides valuable insights, as its results offer opportunities for analysing climatological variability of snow with respect to, e.g., altitudinal zonation.

## Appendix A

The study area belongs to the temperate climate zone. A specific feature of climate in this place is a gradual transition from the oceanic type in the west to the continental one in the east. This is reflected in spatial variability of main climate characteristics such as mean air temperature but also mean snow cover duration (Fig. A1). It ranges from less than 30 days in the northwestern part of the study area to around 100 days in the northeast. The latitudinal pattern is strongly affected by mountainous areas in the south, where snow cover duration is elevation-dependent and can reach 180 days (6 months) in the highest parts of the Carpathians.

Fig. A1. Average snow cover duration (in days) in the period 1991-2020 according to ERA5-Land

The study period (20 winter seasons from 2001/2002 to 2020/2021) is marked by fairly gentle snow conditions with snow duration shorter by around 10 days on average with respect to a long period 1951-2021. It is important to highlight that, in contrast to what is commonly believed, snow conditions then were not yet considerably poorer. By analysing boxplots showing seasonal distribution of snow cover duration and mean snow depth (Fig. A2), it is evident that snow conditions were very variable. The most snowy season regarding snow cover duration was 2005/2006 when 50% of stations had snow for more than 110 days. However, in terms of mean snow depth, this season is slightly inferior to 2004/2005, when median of this parameter reached almost 20 cm. The least snowy season is 2019/2020 with fewer than 10 days with snow at half of the stations and no snow at 25% of them. Mean snow depth was also extremely low then. In general, considering the whole study period, four consecutive seasons 2002/2003 – 2005/2006 as well as 2012/2013 could be regarded as snowy ones, while 2013/2014, 2015/2016 and 2019/2020 are seasons with the poorest snow conditions.

Fig. A2. Seasonal distribution of snow cover duration (a) and the mean snow depth (b) in the study period based on measurements taken at all stations included in the analyses. For the sake of better legibility, outliers are not shown.

Summing it up, average snow conditions in the study period were a little gentler than in the over-70-year period 1951-2022.

However, due to considerable interseasonal variability, extreme winter seasons in the study period are also one of the most

severe and mildest seasons in the long respected long period. This was concluded based on data from ERA5 as well as a number of meteorological stations with long observational series.

Appendix B

Tab. B1. List of reanalysis fields and their aggregations used in training a Random Forests model.

|                                                  |         |           |                       | aggregation/processing                                                                         |  |
|--------------------------------------------------|---------|-----------|-----------------------|------------------------------------------------------------------------------------------------|--|
| parameter name                                   | abbrev. | database  | unit                  | type                                                                                           |  |
| 850 hPa air temperature                          | t       | ERA5      | K                     | daily mean                                                                                     |  |
| surface geopotential                             | z       | ERA5      | $m^2/s^2$             | elevation retrieval                                                                            |  |
| land-sea mask                                    | lsm     | ERA5      | -                     | -                                                                                              |  |
| snow depth                                       | sd      | ERA5      | m of water equivalent | -, snow age                                                                                    |  |
| snow density                                     | rsn     | ERA5      | $kg/m^3$              | -                                                                                              |  |
| 2m air temperature                               | 2t      | ERA5-Land | K                     | 6-hour average, daily average, maximum and minimum, seasonal and weekly sum of T<0°C and T>0°C |  |
| snow albedo                                      | asn     | ERA5-Land | -                     | daily mean                                                                                     |  |
| snow density                                     | rsn     | ERA5-Land | $kg/m^3$              | -                                                                                              |  |
| snow depth                                       | sd      | ERA5-Land | m of water equivalent | -, snow age                                                                                    |  |
| snowfall                                         | sf      | ERA5-Land | m of water equivalent | weekly and seasonal sum                                                                        |  |
| snowmelt                                         | smlt    | ERA5-Land | m of water equivalent | daily and seasonal sum                                                                         |  |
| soil temperature in the uppermost layer (0-7 cm) | stl1    | ERA5-Land | K                     | daily mean                                                                                     |  |
| soil temperature in the second layer (7-28 cm)   | stl2    | ERA5-Land | K                     | daily mean                                                                                     |  |
| surface net solar radiation                      | ssr     | ERA5-Land | $J/m^2$               | daily sum                                                                                      |  |
|                                                  |         |           |                       |                                                                                                |  |

| surface solar radiation  | ssrd  | ERA5-Land | $J/m^2$   | daily average, daily and |
|--------------------------|-------|-----------|-----------|--------------------------|
| downwards                | SSIU  | EKA3-Land |           | seasonal sum             |
| total precipitation      | tp    | ERA5-Land | m         | seasonal sum             |
| snow fraction            | snowc | ERA5-Land | -         | -                        |
| snow depth               | sde   | ERA5-Land | m         | -                        |
| snow density             | rsn   | ERA5-Land | $kg/m^3$  | -                        |
| type of low vegetation   | tv1   | ERA5-Land | -         | -                        |
| type of high vegetation  | tvh   | ERA5-Land | -         | -                        |
| surface geopotential     | Z     | ERA5-Land | $m^2/s^2$ | elevation retrieval      |
| land-sea mask            | lsm   | ERA5-Land | -         | -                        |
| low vegetation fraction  | cvl   | ERA5-Land | -         | -                        |
| high vegetation fraction | cvh   | ERA5-Land | -         | -                        |

Tab. B2. List of DEM-based parameters used in training a Random Forests model.

| parameter name                                 | abbrev. | database | unit              | aggregation/processing type                                                        |
|------------------------------------------------|---------|----------|-------------------|------------------------------------------------------------------------------------|
| potential direct incoming solar radiation      | dirrad  | DEM      | Wh/m <sup>2</sup> | daily and seasonal sum                                                             |
| potential duration of solar radiation          | time    | DEM      | h                 | daily and seasonal sum                                                             |
| elevation                                      | elev    | DEM      | m                 | -                                                                                  |
| distance to the Baltic coast                   | odl     | DEM      | km                | -                                                                                  |
| percentage of surrounding area elevated higher | major   | DEM      | %                 | for a set of radii [m]: 100,<br>200, 500, 1000, 2000, 5000                         |
| sky view factor                                | SVF     | DEM      | -                 | for a set of radii [m]: 100,<br>200, 500, 1000, 2000, 5000                         |
| topographic position index                     | tpi     | DEM      | -                 | for a set of radii [m]: 100,<br>200, 500, 1000, 2000, 5000,<br>10000, 20000, 30000 |
| date                                           | date    | -        | -                 | -                                                                                  |
| day of a season                                | doy     | -        | -                 | starting from 1st November                                                         |
| month of a season                              | moy     | -        | -                 | starting from November                                                             |
| year                                           | year    | -        | -                 | -<br>-                                                                             |
| station rank                                   | rank    | -        | -                 | encoded to integer                                                                 |
| longitude                                      | lon     | -        | 0                 | -                                                                                  |
| latitude                                       | lat     | -        | 0                 | =                                                                                  |

## Data availability

Snow depth measurements from Polish stations used in the study are publicly available at danepubliczne.imgw.pl (accessed: 530 23.02.2025). This does not include data from precipitation stations from years 2001-2010, which are available only internally. https://doi.org/10.5194/egusphere-2025-5084 Preprint. Discussion started: 24 November 2025 © Author(s) 2025. CC BY 4.0 License.

For downloading the data, an R package *climate* was used (Czernecki et al. 2019). As far as Czech stations are considered, data are publicly available at https://www.chmi.cz/historicka-data/pocasi/denni-data (accessed: 14.02.2024). Data from Slovakia are available only on request at the national weather service (Slovenský hydrometeorologický ústav).

Copernicus reanalyses ERA5 and ERA5-Land are available to download at the Climate Data Store of Copernicus Climate

Change Service at https://cds.climate.copernicus.eu/datasets (accessed: 23.06.2024)

#### **Author contributions**

GS and ZU conceptualized the goals of the research; GS downloaded and preprocessed the data. The development of methodology and training of ML models was performed by GS. Investigations and validation were done by GS under the supervision of ZU. All figures were made by GS. ZU prepared Appendix A, GS prepared the rest of the manuscript. ZU reviewed and proposed minor corrections to the final version of the manuscript.

#### Competing interests

The authors declare that they have no conflict of interest.

#### Acknowledgements

The measurement data from Slovakian stations come from the Slovakian National Weather Service (Slovenský hydrometeorologický ústav) and were provided thanks to the kind support of Jozef Pecho. Computational environment was provided by high-performance Infrastructure PLGrid (HPC Centers: ACK Cyfronet AGH, PCSS, CI TASK, WCSS).

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
