# Peer review of "Identification and correction of snow depth bias in ERA5 datasets over Central Europe using machine learning"

_EGUsphere, 2025_

## Author Comment (AC1)

**REVIEWER 1**

**MAJOR COMMENTS:**

1.  ***Justification and contribution***: *The manuscript would benefit from a stronger emphasis on the novelty and significance of the work. At present, the motivation and main contributions are not highlighted clearly enough. What specific gap does this study address? What are the main contributions to the field? These points should be explicitly stated in the Introduction and reinforced in the Discussion and Conclusions.*

    Answer:

    Thank you for this relevant suggestion. The paragraph concerning the goal of the study have been reformulated so that the novelty of the work is emphasized in the Introduction, as well as in the Conclusions.

2.  ***Broader perspective***: *How does this study contribute to snow depth estimation in a broader context? Could the proposed approach be applied to other regions or datasets? What are the future perspectives for this line of research? Addressing these questions would enhance the impact and relevance of the work.*

    Answer:

    The questions have been addressed at the end of the Conclusions section.

**MINOR COMMENTS:**

1.  *Why not use snow depth (sde) from ERA5-Land directly?*

    Answer:

    Thank you for this very relevant question. At the initial state of research, we asked ourselves the same question. We compared  seasonal variability of both fields at every station and it turned out that they are virtually identical. The Pearson's coefficient between them is close to 1 (0,999...) and maximum differences does not exceed 1-2 cm. These results are valid for all types of stations, regardless inclusion into the data assimilation system. Therefore, we believe that the direct snow depth field available in ERA5-Land was calculated in the same way as it was done in our research. Rare differences that occur seem to have purely numerical origin. In a plot below (Fig. 1) you can see an example of seasonal snow depth variability for these two fields.  The two lines match each other very closely and there are only a few days with small ( <=1 cm) discrepancies.

[Figure]

*Fig. 1 Snow depth variability represented by direct snow depth field from ERA5-Land (sde) and diagnostic field derived from SWE and snow density (sd_diag) at station Svratouch in the 2009/2010 season.*

2. *Consider replacing "interseasonal" with interannual.*

   Answer:
   The suggestion have been incorporated throughout the manuscript.

3. *References are missing in L29–32 and L40.*

   Answer:
   The references have been added.

4. *L11: Consider changing "database" to datasets.*

   Answer:

   The phrase has been modified as suggested.

5. *L19: The phrase "lower-ranked stations" is unclear, as the ranking is explained only later in the Introduction. Consider rephrasing.*

   Answer:
   Thank you for pointing this out. After revision, station ranks are named explicitly (climate and precipitation stations).

6. *L64: Clarify what is meant by "ERA5 regional ... reanalyses."*

   Answer:
   Thank you for this comment, the sentence is indeed unclear. What we meant was "global and regional reanalyses other than ERA5". Th sentence have been rephrased to: "(...) concerning ERA5 as well as other regional and global reanalyses (..)"

7. *L68: Spell out WRF on first use.*

   Answer:

The abbreviation has been expanded.

8.  *L76: Spell out ML on first use.*

    Answer:
    The abbreviation has been introduced at the first usage of "machine learning", which is five lines above the indicated issue.

9.  *L77: Provide full names for MERRA-2, GLDAS, CMC*

    Answer:
    The abbreviations have been expanded.

10. *L87–88: The sentence "A limited amount of research in this region regard also ML-based bias correction of snow estimation in reanalysis" is difficult to understand. Please reword for clarity.*

    Answer:

    The sentence has been removed due to revision of the paragraph concerning the goals of the study.

---

## Author Comment (AC2)

**REVIEWER 2**

COMMENTS:

1. *L202 Can you comment if 70 predictors are in normal range for this type of models?*

   Answer:

   The number is more than usually reported in literature. Typically, there are around 10-20 features. These are mostly a few basic meteorological variables (air temperature, precipitation) and a set of environmental variables (related to topography and vegetation). A great majority of features occurring in literature is included in our model. There are a few reasons that our dataset is bigger. At first, predictions of our RF model are time independent and therefore, some variables have been aggregated in different time intervals. Also, temporal variables had to bee added, which is not the case when an LSTM is run (e.g., Cui et al., 2023). Secondly, some input variables are highly correlated, despite their significant score in variable importance metric. These are for example: snow density and snow albedo, 850hPa air temperature and 2-m air temperature. Last but not least, some variables have to be doubled (despite their high collinearity) since we take two reanalyses on input (e.g., snow depth, model elevation). An interesting approach was adopted by Tanniru et al. (2025) – they initially prepared 40 predictors, however, they excluded those which were highly correlated (R>0.80), so they ended up running a model with 16 predictors.

   A short explanation has been added in Section 2.5.

2. *L204 Why was number of trees set to 100?*

   Answer:

   You must have mistyped the number of trees as in the paper stands 1000, not 100. The parameter is equal by default to 500 and was later doubled as more and more predictors where included into training. Boehmke and Greenwell (2019) suggest it should be around 10 times their number. Additionally, increasing the number of trees ensures more stable estimation of the out-of-bag error and variable importance (at the expense of longer training time though). A short justification was added in the manuscript.

3. *L214 Can you elaborate bit more why no spatial split of data was tested? Even for just few years? It would be interesting to see how model performs in new locations.*

   Answer:

   Thank your for this valuable suggestion! The training strategy has been extended so that, beside temporal cross-validation, also spatial split is considered. Fur this purpose, the study area has been split into five longitudinal blocks (2° each). Such a split ensures that every block includes both lowland stations with rare snow occurrence as well as elevated stations with deep snow cover. Then, cross-validation was performed with 4 blocks being a training set, and the remaining one – a test set. The variability of the RMSE training error across different combinations is limited which confirms solid generalization skill of the RF model. Section 2.5 has been updated on the above mentioned information.

4. *Figure 2 The caption text is little confusing, maybe clarify that left side are the input predictors and right is the target (station data) used for training.*

   Answer:

   The caption has been corrected according to Your suggestion.

5. *L 248 I recommend adding subsections to the result section to make it easier to follow.*

Answer:

The section have been split into four subsections.

6. *L248-263 This paragraph is quite long, considered dividing into two parts*

Answer:

The suggested change have been introduced.

7. *L260 What about station near the sea level but away from the shore?*

Answer:

In order to examine it, a plot of MAE as a function of elevation was constructed considering only stations located below 150 m a.s.l. (see Fig. 2 below). Additionally, distance to the sea coastline (in km) is indicated for each station.  In fact, there are only a few stations situated below 50 m a.s.l. that are not at the coast. An increase of MAE for coastal stations is clearly evident if only synops are considered (red dots).  In case of lower-rank stations, two climate stations (blue dots) exhibit an increased MAE, one of them lies just 2 m a.s.l. but 18 km away from the coastline. However,  their MAE values do not stand out that much from other, more inland climate stations.  Consequently, there is not enough evidence to conclude that stations lying near sea level but away from the coastline have increased errors.

A minor change have been made in the manuscript to emphasize that this concerns mostly synop stations.

[Figure]

*Fig. 1 MAE of snow depth in relation to elevation in ERA5 for stations lying not higher than 150 m a.s.l. Distance to the Baltic coast is showed as a label next to every point (in km). Stations are coloured according to their rank (red - synop, blue - climate, white – precipitation).*

8. *Figure 7, Quite hard to see different years in figure (even bolded ones)*

Answer:

The figure has been significantly modified. Instead of spaghetti plot, a ribbon (plume) plot have been used so the readability is enhanced.

9. *L312 Might be good idea to remind reader about the resolutions of the grids here (or in figure 8 caption)*

Answer:

Thank you for this relevant suggestion. Information about horizontal resolution has been added in brackets in the indicated sentence as well as in the preceding one.

10. *L315 Can you clarify if data from the stations used to validate downscaling was also used for training model?*

Answer:

The training dataset does include data from these stations, however, not from this winter season. The predictions were generated through the strategy of temporal split (19 seasons of training, one for testing). We laid particular stress to the fact that a test set includes solely unseen data. We can assure you that there is no data leakage.

11. *L424 There are two commas after "all"*

Answer:

The redundant comma has been removed.

12. *L455 While it's clear to most readers what "this part of Ventral Europe" means, might be good idea to be bit more precise here*

Answer:

Thank you for this relevant suggestion! We have added a directional term "north-eastern" to be more precise.

REFERENCES:

Boehmke, B. and Greenwell, B.: Hands-On Machine Learning with R, 1st ed., Chapman and Hall/CRC, https://doi.org/10.1201/9780367816377, 2019.

Cui, G., Anderson, M., and Bales, R.: Mapping of snow water equivalent by a deep-learning model assimilating snow observations, Journal of Hydrology, 616, 128835, https://doi.org/10.1016/j.jhydrol.2022.128835, 2023.

Tanniru, S., Singh, D. K., Singh, K. K., and Ramsankaran, R.: Exploring Machine Learning's Potential for Estimating High Resolution Daily Snow Depth in Western Himalaya Using Passive Microwave Remote Sensing Data Sets, Earth and Space Science, 12, e2024EA003849, https://doi.org/10.1029/2024EA003849, 2025.

---

## Author Comment (AC3)

**REVIEWER 3**

**MAJOR COMMENTS:**

1. *One major missing element from the manuscript is the discussion of elevation mismatch between the coarse reanalysis gridcells and stations. Both in terms of evaluation, because from all previous studies it emerges that if one accounts for these differences, errors drop considerably. But also for the RF, it could be a key input variable.*

   Answer:

   Elevation mismatch is certainly the primary factor contributing to systematic error of snow depth over complex orography. If in Fig. 4a-b, the mean bias is plotted against the elevation difference instead of absolute elevation value, there will be a distinct linear relationship between them (see Fig. 3 below). However, bias changes during the season (increases with snow accumulation), so trend parameters derived from the plots would not be relevant to calculate a daily correction. Hence, apart from elevation mismatch, other factors (e.g., absolute value of snow depth, accumulated precipitation since the beginning of the season) should also be explanatory. It is definitely more common in literature to account on elevation mismatch when correcting bias of air temperature using lapse rate since the method is very simple and the parameter is instantaneous (Bouallègue et al., 2023; Keller et al., 2021). When it comes to snow fields, it is common by dynamical downscaling to apply lapse-rate-based corrections to fields that determinates snow at most, i.e. air temperature and precipitation (Baba et al., 2021; Dalla Torre et al., 2024). However, we are not aware of any publication where a snow field would be corrected using some linear relation based on elevation mismatch. We can imagine that it could possibly reduce the error in complex terrain to some extent, however, please notice that even for stations with little elevation difference, there are still non-negligible systematic errors. An example of such site is Puczniew (Fig. 4), a climate station which lies in central Poland in nearly flat terrain, with only 3 metres of elevation difference against ERA5 and a perfect match in case of ERA5-Land. The station lies around 20 km away from a synop station which probably was assimilated and therefore the mean bias for the atmospheric reanalysis is relatively low. However, RF is still able to make it better.

   Summing it up, the approach we used is to tackle the systematic error in total, regardless weather it was predominantly driven by elevation mismatch, simplifications in parametrizations of snow physical processes or any other factor (although some part of the error could be reduced with some simple linear method). Such an approach is common when correcting snow bias using ML methods.

[Figure]

*Fig. 1 Mean bias of snow depth for every station in relation to elevation mismatch (reanalysis minus real) for ERA5-Land (right) and the atmoshperic ERA5 (left).*

[Figure]

*Fig. 2 Snow depth variability in the 2005/2006 season at station Puczniew. Despite negligible elevation mismatch, systematic errors still exist.*

Regarding the inclusion of elevation mismatch at the stage of a RF model training – this piece of information is indirectly provided to the model with three variables: elevation from the Digital Elevation Model (as a proxy of real elevation) and model elevation of ERA5 and ERA5-Land. If we used relative fields instead (difference instead of absolute values), they would probably be the most important features during training. However, some information regarding absolute altitude might be lost. Actually, we conducted such an experiment and the difference in training MSE is negligibly small (6.321 versus 6.315 cm). Thus, the two alternative forms of information about elevation could be considered as equivalent.

2. *In the RF date, day, month, and year are used as input. In an operational setting, year and date would not be available? From the variable importance analysis, they seem to have some influence. Would it make sense to test a model without these variables?*

Answer:

We cannot see any reason why temporal variables like day, month, year and date would not be available when running operationally. However, it was not the goal of this study to propose a tool that could be run operationally. The main reason for it is that ERA5 reanalyses are publicly available with a delay of around 5 days.

3. *Sec 2.5 unclear how you split into training, test and validation sets. Was a validation set used at all? Similar to the previous reviewers, I strongly suggest including a validation set in the spatial domain. Moreover, it could be useful to give summary metrics for the different sets (training, test, validation), to see how well the model generalizes.*

Answer:

There is no separate validation set in a sense that occurs by other machine learning methods, e.g. artificial neural networks. In contrast to them, random forests (RF) does not use a separate, fixed validation dataset due to different training methodology. RF is a decision-trees-based method that uses bootstrap aggregating – a resampling method that randomly generates multiple data samples with some data replaced by duplicated samples of the original set (Boehmke and Greenwell, 2019). The part of the original dataset not included in a bootstrap sample (around 27% on average) is considered out-of-bag (OOB). The OOB samples are then used to validate the model. Hence, the score is called the OOB error. Breimann (2001) explicitly stated that the error "*removes the need for a set aside test set*". Under "test set" he meant "validation set" – these two terms used to be often confused in literature (Ripley, 1996). The OOB error is proved to be a good estimation of the model generalisation error (Breiman, 2001). Hence, the error is commonly regarded as a training error, while *sensu stricto* this is a validation error.

Furthermore, we appreciate your suggestion regarding the spatial split. The training strategy has been extended so that, beside temporal split, also spatial split is concerned. Taking into consideration latitudinal pattern of land relief in the study area, the domain has been split into 5 longitudinal bands (every 2°), so that every band includes some mountainous stations which provide extensive data. Little variability of training error in both temporal and spatial split proves good generalization skill of the RF model. A detailed list of the errors for every combination is put in the Appendix C. In the manuscript (section 2.5), the information was given in an aggregated form (mean + standard deviation). In addition, the description of splitting was put in a separate paragraph in the Section 2.5 so that it is more comprehensible.

4. *Example downscaling: it unclear if the authors used interpolation of surface meteorology from stations using bicubic? Or what variables were interpolated to perform the downscaling? Note that simple bicubic is not appropriate for variables that have a strong elevation dependency such as temperature, humidity, … I don't know if this might be an explanation for the errors found. Of course it is difficult to validate such a dataset, but have you considered remote sensing products based on MODIS, such as globalsnowpack from the DLR or ESA snow CCI? Of course you'd have to convert snow depth to snow presence, but it could give you some independent spatial information.*

Answer:

Thank you for this question. None of the station-measured parameters have been interpolated. Bicubic interpolation was performed over predictors from reanalyses in order to prepare a test set. As they already were continuous 2D fields before this operation, it was actually regridding rather then interpolation. We are aware that some of the fields are meteorological fields with distinct elevation dependence which is not accounted for while interpolating bicubicly. However, please notice that during retrieval of a point value from a reanalysis (as it was done for every station in the study area for training), it is often interpolation from the nearest node which is performed (alternatively, it could be a raw value from the nearest node, but in our case it wouldn't make sense). Therefore, the predictors are not really downscaled, but rather bicubically regridded. The description of the experiment setup in Section 2.6 has been modified so that it is hopefully more accurate and clear.

Regarding using an independent dataset, we are very grateful for this suggestion. At the initial stage of research, we found the remote-sensing-based products not relevant to our research as they mostly provide qualitative information about snow, not quantitative. However, it is indisputable that in cases where snow does not cover the entire study area, information about snow presence do provides added value. In order to fully benefit from it, date of the presented results had to be changed (at the initial date, the whole domain was covered by snow). Consequently, major modifications have been introduced in Section 2.4 and 3.3, including the figures. The snow mask from the GlobSnowPack database was added upon Fig. 8b.

MINOR COMMENTS:

1. *L69: Avanzi and Fontrodona are not appropriate references for the statement.*

   Answer:

   Thank you for pointing it out. Indeed the work of Avanzi et al. is an example of reanalysis that does not involve any numerical modelling. The second study we referred to does employ calculation using a regional snow model ΔSNOW, however only point-wise (for stations). Therefore, it cannot be considered dynamical downscaling. The incorrect references have been replaced with the relevant ones.

2. *L90: I guess topographic complexity can also be high in the Americas and HMA, depending on where you are.*

   Answer:

   What we intended to convey is that spatial resolution of the output of the ML models proposed in referred papers (specifically, Cui et al. (2023), King et al. (2020), Tanniru (2025)) is still insufficient regarding the scale of topographic complexity that occurs in the highest mountain range in our study area. No doubt that such complexity occurs also in mountains of North America or in the Himalayas. However, the papers concerning these regions propose spatial resolution that is indeed finer that the output, but with primary goal to accurately reflect snow depth (or SWE) over a large mountainous area, rather than offering horizontal resolution corresponding to the scale of topographic complexity. The sentence has been reformulated for clarity.

**REFERENCES**

Baba, M. W., Boudhar, A., Gascoin, S., Hanich, L., Marchane, A., and Chehbouni, A.: Assessment of MERRA-2 and ERA5 to Model the Snow Water Equivalent in the High Atlas (1981–2019), Water, 13, 890, https://doi.org/10.3390/w13070890, 2021.

Boehmke, B. and Greenwell, B.: Hands-On Machine Learning with R, 1st ed., Chapman and Hall/CRC, https://doi.org/10.1201/9780367816377, 2019.

Bouallègue, Z. B., Cooper, F., Chantry, M., Düben, P., Bechtold, P., and Sandu, I.: Statistical Modelling of 2m Temperature and 10m Wind Speed Forecast Errors, Mon. Wea. Rev., 1, https://doi.org/10.1175/MWR-D-22-0107.1, 2023.

Breiman, L.: Random Forests, Machine Learning, 45, 5–32, https://doi.org/10.1023/A:1010933404324, 2001.

Cui, G., Anderson, M., and Bales, R.: Mapping of snow water equivalent by a deep-learning model assimilating snow observations, Journal of Hydrology, 616, 128835, https://doi.org/10.1016/j.jhydrol.2022.128835, 2023.

Dalla Torre, D., Di Marco, N., Menapace, A., Avesani, D., Righetti, M., and Majone, B.: Suitability of ERA5-Land reanalysis dataset for hydrological modelling in the Alpine region, Journal of Hydrology: Regional Studies, 52, 101718, https://doi.org/10.1016/j.ejrh.2024.101718, 2024.

Keller, R., Rajczak, J., Bhend, J., Spirig, C., Hemri, S., Liniger, M. A., and Wernli, H.: Seamless Multimodel Postprocessing for Air Temperature Forecasts in Complex Topography, Wea. Forecasting, 36, 1031–1042, https://doi.org/10.1175/WAF-D-20-0141.1, 2021.

King, F., Erler, A. R., Frey, S. K., and Fletcher, C. G.: Application of machine learning techniques for regional bias correction of snow water equivalent estimates in Ontario, Canada, Hydrology and Earth System Sciences, 24, 4887–4902, https://doi.org/10.5194/hess-24-4887-2020, 2020.

Ripley, B. D.: Pattern Recognition and Neural Networks, 1st ed., Cambridge University Press, https://doi.org/10.1017/CBO9780511812651, 1996.

Tanniru, S., Singh, K., Singh, K., and Ramsankaran, R.: Exploring Machine Learning's Potential for Estimating High Resolution Daily Snow Depth in Western Himalaya Using Passive Microwave Remote Sensing Data Sets, 2025.